# A Visual SLAM Robust against Dynamic Objects Based on Hybrid Semantic-Geometry Information

**Sheng Miao [1], Xiaoxiong Liu [2],[*], Dazheng Wei [1] and Changze Li [1]**

1    School of Automation, Northwestern Polytechnical University, Xi'an 710072, China;
     2015303413@mail.nwpu.edu.cn (S.M.); phosgene@mail.nwpu.edu.cn (D.W.); cz_li@mail.nwpu.edu.cn (C.L.)
2    Shaanxi Province Key Laboratory of Flight Control and Simulation Technology, Xi'an 710072, China
*    Correspondence: liuxiaoxiong@nwpu.edu.cn

**Abstract:** A visual localization approach for dynamic objects based on hybrid semantic-geometry information is presented. Due to the interference of moving objects in the real environment, the traditional simultaneous localization and mapping (SLAM) system can be corrupted. To address this problem, we propose a method for static/dynamic image segmentation that leverages semantic and geometric modules, including optical flow residual clustering, epipolar constraint checks, semantic segmentation, and outlier elimination. We integrated the proposed approach into the state-of-the-art ORB-SLAM2 and evaluated its performance on both public datasets and a quadcopter platform. Experimental results demonstrated that the root-mean-square error of the absolute trajectory error improved, on average, by 93.63% in highly dynamic benchmarks when compared with ORB-SLAM2. Thus, the proposed method can improve the performance of state-of-the-art SLAM systems in challenging scenarios.

**Keywords:** SLAM; optical flow; image segmentation; dynamic objects

## 1. Introduction

Pose estimation, a key branch of image processing, is expected to play an increasingly important role in key future technologies such as augmented reality, unmanned driving, and human–robot interactions. However, the Global Positioning System (GPS) [1], a common navigation system, often fails to function correctly indoors or in unknown and complicated environments. Fortunately, with the rapid development of computer vision in recent years, researchers are able to use cameras as external sensors to achieve accurate positioning in complex environments [2,3]. Due to their light weight and wide application, cameras represent a popular means of position estimation on drones and robots. By comparing changes in the images captured via a camera, robots can calculate the position transformation matrix and reconstruct the real-world map, a process referred to as visual simultaneous localization and mapping (SLAM).

Generally, the visual SLAM framework consists of four parts: the front-end visual odometer, back-end optimization, a loop detection module, and a mapping module. These frameworks are considered to be either feature-based or direct methods. The feature-based method minimizes the reprojection error between two images, while the direct method minimizes the photometric error under the assumption of constant gray level. Over recent years, there have been some outstanding works in the field of SLAM such as Mono-SLAM [4], ORB-SLAM2 [5], LSD-SLAM [6], DVO [7], SVO [8], and so forth. For example, Mono-SLAM [4] is the first feature-based SLAM system based on the extended Kalman filter. While this system is initially capable of creating sparse maps online, the effect is not ideal due to the large amount of drift. On the other hand, LSD-SLAM represents the first large-scale direct method. LSD-SLAM [6] maintains highly accurate pose estimation based on direct image alignment, as well as reconstruction of the 3D environment as a pose-graph of keyframes with associated semi-dense depth maps. However, this system is very sensitive

to the influence of exposure and the camera internal matrix. ORB-SLAM2 [5], proposed in 2015, is a sophisticated system that boasts map reuse, loop closing, and relocalization capabilities. This system functions well in a wide range of situations—from small handheld drones flying indoors to larger drones flying in complex outdoor environments—and achieves state-of-the-art performance. The accuracy of the pose estimation achieved by these systems in static environments is excellent.

However, some pertinent issues remain unsolved. As most available SLAM systems rely on strong assumptions regarding static environments, they are less applicable to dynamic scenarios—thus preventing their deployment in real-world situations. The associated dynamic elements violate these static environment assumptions and can lead to failures in the tracking process. Moreover, dynamic elements are fused into the scene map if they are not actively eliminated, which leads to system corruption. In the majority of SLAM systems, dynamic keypoints are considered as outliers and thus should be excluded from the map [9–11]. Along with the guidance, there exist some representative dynamic SLAM systems from recent years that can also deal appropriately with the content of dynamic scenes. Such systems can be divided into two main types: those which are geometry-based, and those which are learning-based.

The geometry strategy that is most commonly applied is to treat dynamic moving elements as noise that must be distinguished and eliminated. Geometry-based approaches do not require prior information of moving objects, and thus their processing speed is faster when compared with that of learning-based methods. Zhang et al. [12] proposed FlowFusion, a system that uses a learning-based network to obtain the predicted optical flow of each frame before synthesizing a wrapped image after removing the camera motion through the use of a visual odometer. The segmentation area of dynamic objects can then be obtained, and static backgrounds reconstructed. Cheng et al. [13] proposed an optical-flow-based approach which uses optical flow to distinguish and eliminate dynamic feature points from those extracted using RGB images as the sole input. The camera ego-motion—estimated by the essential matrix of two consecutive frames—is utilized for optical flow computation in their system. StaticFusion [14] leverages a probabilistic segmentation of the image to reconstruct the background, and integrates it into a weighted dense optimization framework. Li et al. [15] proposed a static weighting method for edge points in the keyframe. By calculating the likelihood of each keyframe point being part of the static environment, this system significantly reduces the interference of dynamic objects. More recently, based on the fact that static objects should exhibit continuous pose transformation over time, Dai et al. [16] exploited point correlations to separate static and dynamic map points. Although the geometry-based method improves the robustness of SLAM systems to a certain extent, additional effort to improve the accuracy of the pose estimation is required, as it has room for further improvement when compared with the learning-based method.

Numerous learning-based approaches have been investigated. Yu et al. [17] proposed DS-SLAM on the basis of ORB-SLAM2. This system combined the SegNet network with a motion consistency-checking method to reduce the influence of dynamic objects. DS-SLAM retains high accuracy in highly dynamic environments. The Dyna-SLAM [18] system is capable of detecting moving objects via multi-view geometry, Mask R-CNN, or both. This system detects dynamic objects and produces a static map of the real-world environment. On this basis, the input frame background is repainted to fill in the area concealed by dynamic objects. Detect-SLAM [19] incorporates a deep neural network (DNN) object detector into the SLAM system to benefit from these two mutually beneficial functions. This system categorizes keypoints into four states according to their moving probability, and then removes all points with a high moving probability to maintain a robust pose estimation. Lv et al. [20] proposed a semantic flow-guided motion removal method which consists of four deep learning networks: depth network, pose network, flow network, and semantic network. By leveraging semantic information and optical flow to extract motion regions, it achieves perfect performance in complicated environments.

More recently, DP-SLAM [21] combines epipolar constraints and semantic segmentation for optimization within a Bayesian probability framework. As the front-end of the ORB-SLAM2, this method significantly improves the precision of the state-of-the-art SLAM system in various challenging scenarios. This learning-based method can identify dynamic objects without the need of multi-frame processing. However, since convolutional neural networks [22–24] are trained using public datasets and only limited categories of objects can be determined, some dynamic keypoints can be mistakenly defined as static, thus contaminating the background model and triggering system corruption.

In this paper, a robust hybrid SLAM system is proposed, which involves two convolutional neural networks to not only impede the interference of dynamic objects, but to also compute the movement velocity. The main contributions of this paper are summarized as follows:

(1) We propose a hybrid RGB-D SLAM that features two advanced deep learning networks: DeepLabv3 for semantic segmentation, and PWC-Net for optical flow prediction. The hybrid SLAM is capable of curbing the interference of dynamic elements in complicated scenes via the combination of geometry and semantic modules, while retaining sufficient static elements for accurate estimation of position.

(2) We propose an efficient optimization strategy based on geometric relationships to synthesize the coarse moving area of the current frame. Leveraging the depth image and the matched keypoints, we employed bundle adjustment to calculate the initial transform matrix and smooth the crude optical flow generated by PWC-Net. Then, we applied the k-means algorithm to cluster the region with relatively high flow value, and supplemented this via an epipolar check. Additionally, as a technical implementation of the optical flow residuals, we provide the speed of the common dynamic instance in our mechanism, especially in reference to the person studied in the present paper.

(3) To verify the effectiveness of the proposed method, we performed a systematic evaluation of both the benchmark sequences and datasets recorded by our experimental quadcopter. Compared with state-of-the-art SLAM (e.g., ORBSLAM2) and other prominent dynamic SLAM systems, our approach demonstrated superior intelligence in challenging scenarios.

## 2. Overview of the Framework

It is seldom the case that robots operate in absolutely static scenarios, and such a requirement would significantly confine the extent to which they could be successfully deployed. In this overview of our work, we propose a robust hybrid RGBD-SLAM that harmonizes semantic association and optical flow–geometric constraints to address this issue. Firstly, two adjacent frames are taken as input to the PWC-Net [25] and Deeplabv3 networks [26] to generate a full optical flow and a semantic mask. Then, we smooth the full optical flow through the Lightly Track module and apply k-means to cluster the static and dynamic points. After a supplemental epipolar check, the clustered dynamic points are fed into the specific cost function to create a geometry mask. Finally, our approach synthesizes the precise moving area, leveraging the cooperation of the two modules. This system can also provide the velocity of human movement as a technical implementation of the optical flow. The modified SLAM system integrating our proposed motion removal method is built on ORB-SLAM2. A flowchart of the proposed approach is presented in Figure 1.

To implement the SLAM system in dynamic environments, we built a quadcopter (see Figure 2) to record the official datasets. The main components of the platform included a Jetson TX2 (Nvidia, Santa Clara, CA, USA), a flight control unit (FCU), and an MYNT EYE D1010-IR-120 (MYNT AI, Wuxi Shi, Jiangsu, China). We employed a NVIDIA TX2 as the image-processing unit—running the Ubuntu18.04 OS—and selected STM32F427 as the FCU. Communication between the FCU and the TX2 was carried out through the serial port using the ROS [27] package. The onboard camera, an MYNT EYE D1010-IR-120 (MYNT AI, Wuxi Shi, Jiangsu, China), provided the RGB and depth images in real time.

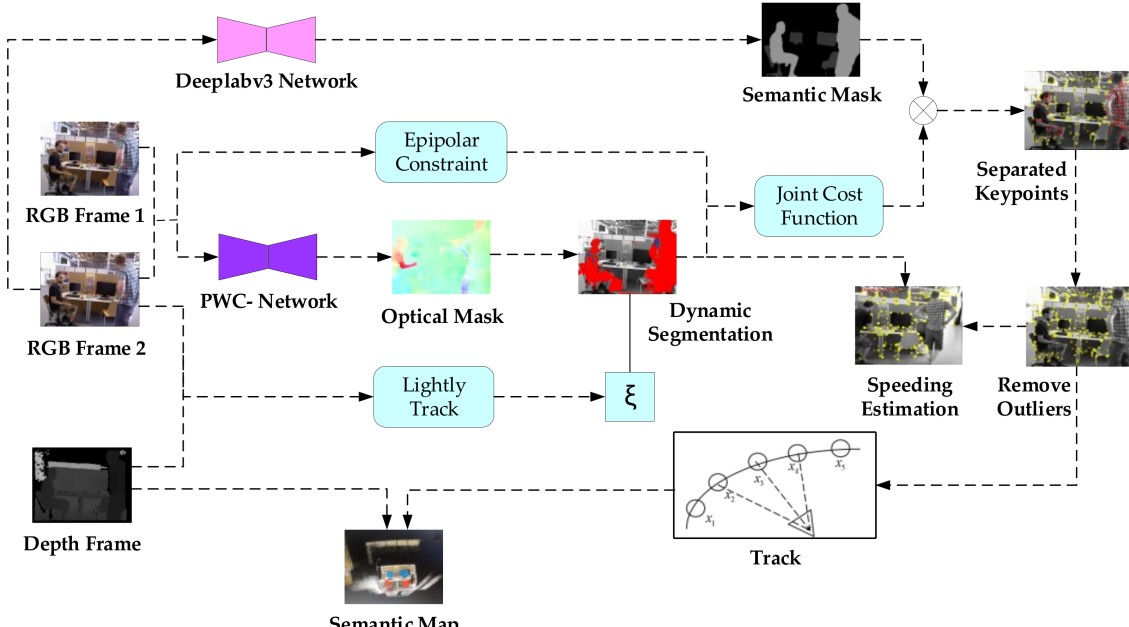

**Figure 1.** Block diagram of the proposed approach. In our pipeline, two raw adjacent RGB frames and a depth frame are taken as input. The system is composed of semantic segmentation and geometry segmentation modules. The images pass through Deeplabv3 to compute the semantic mask and distinguish physical coarse dynamic regions in the geometry pipeline before the separated keypoints can be obtained via cooperation between the two modules.

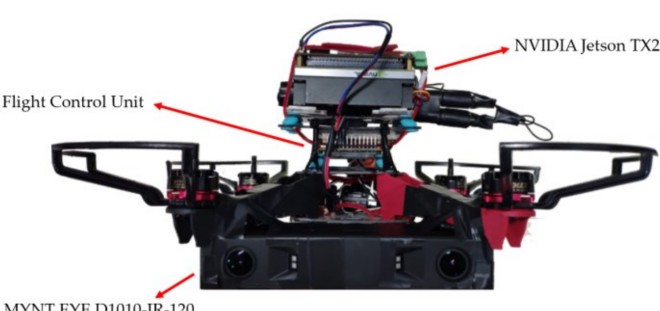

**Figure 2.** The experimental quadcopter.

## 3. Methodology

In this section, we elaborate how we combined geometric and semantic information to detect moving regions in dynamic environments. The framework is primarily composed of three parts: the geometric module, the semantic segmentation module, and the outliers rejection algorithm.

### 3.1. Optical Flow Residual Clustering

Optical flow [28,29] is a very effective method for segmenting moving objects, compared with other motion segmentation methods, due to its low sensitivity to light and irrelevant events. However, if the unmodified method alone is utilized to obtain moving object optical flow, it will fail to provide accurate data as the rudimentary optical flow includes both the movement of the static background—caused by the motion of the camera—and the movement of the dynamic object itself. Given two adjacent frames A and B, we introduce the concept of 2D full flow $\delta x_{A \rightarrow B}^{ff}$ to describe the rudimentary flow, which can be easily obtained from the paired images. We define the motion of the static background as rigid

flow $\delta x^{rf}_{A \to B}$. Then, the projected 2D optical flow residuals $\delta x^{of}_{A \to B}$ [30,31] on the image plane (see Figure 3) can be computed as:

$$\delta x^{of}_{A \to B} = \delta x^{ff}_{A \to B} - \delta x^{rf}_{A \to B}. \tag{1}$$

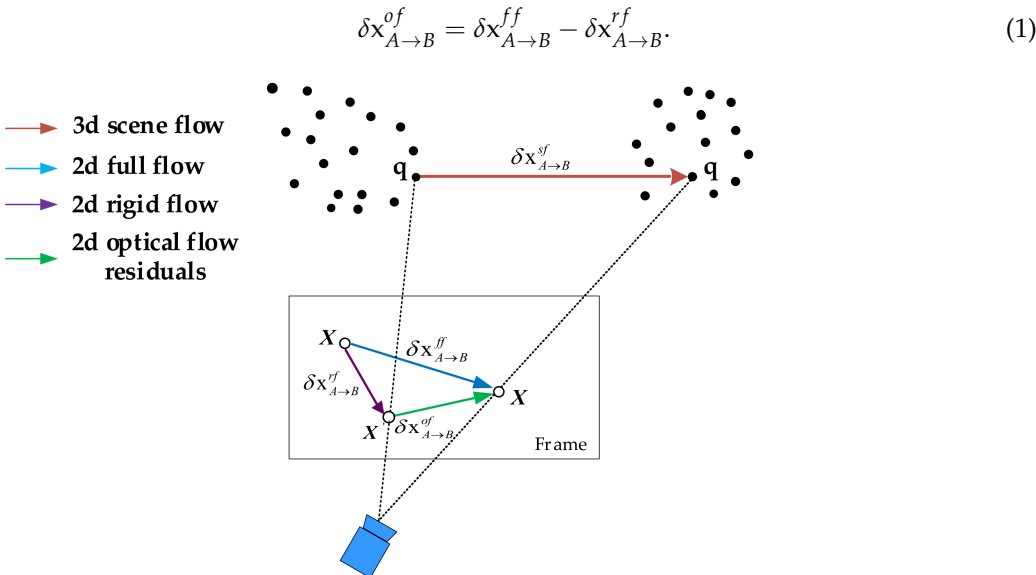

**Figure 3.** The projected 2D optical flow residuals in the image planes: q is the 3D point in world coordinate; the red arrow indicates the 3D scene flow, which is the world coordinate motion; the blue arrow is the 2D full optical flow obtained via PWC-Net; the purple arrow is the 2D rigid flow resulting from camera ego-motions; and the green arrow indicates the 2D optical flow residuals triggered by the dynamic objects.

To remove the camera ego-motion contamination, we denote $\xi \in \mathfrak{se}(3)$ as the rigid motion of the camera, which can be solved via the bundle adjustment algorithm:

$$\xi = \underset{\xi}{\arg\min} \sum_{i=1}^{n} \left\| u_i - f_{proj}(P_i, \xi_i) \right\|_{\Sigma}^{2}, \tag{2}$$

where $\xi$ represents the camera pose matrix and $u_i$ represents the 2D point on the image plane. Through iteratively adjusting the pose $\xi$ of the camera to minimize the L2 norm of the reprojection error between the projected 2D point $f_{proj}$ and the corresponding keypoint $u_i$, we can acquire the optimized camera's 6 degrees of freedom (DOF) motion. In our pipeline, we designed the Lightly Track module, where the EPNP algorithm [32] provides an initial motion guess and the low-cost bundle adjustment further optimizes the initial motion. For one 2D pixel $x$ of frame A, the rigid flow can be computed as:

$$\delta x^{rf}_{A \to B} = \Psi(x, \xi) - x. \tag{3}$$

The $\Psi$ stands for an image warping operation:

$$\Psi(x, \xi) = \pi \left( T(\xi) \pi^{-1}(x, D(x)) \right). \tag{4}$$

The transformation matrix is defined as $T(\xi) \in SE(3)$, which is a transformation of $\xi \in \mathfrak{se}(3)$. $D(\cdot)$ indicates the depth value of the pixel $x$ and $\pi$ represents the projection operation from world coordinate to camera pixel:

$$\pi : \mathbb{R}^3 \to \mathbb{R}^2. \tag{5}$$

For the background, Equation (1) is close to zero—since its full flow comes from the camera motion—whereas the dynamic pixels are not zero since the full flow consists of both

camera and object motion. Thus, we apply the k-means algorithm to separate the dynamic region from the static background. We denote the two cluster sets as $C_1 = (p_{d1}, p_{d2}, \ldots, p_{dn})$ and $C_2 = (p_{s1}, p_{s2}, \ldots, p_{sn})$. Moreover, we can easily obtain the 3D scene flow $\delta x_{A \to B}^{sf}$ under real-world conditions. We characterize $\delta x_{A \to B}^{sf}$ by the following formula:

$$\delta x_{A \to B}^{sf} = \pi^{-1}\left(x + \delta x_{A \to B}^{of}, D_B\left(x + \delta x_{A \to B}^{of}\right)\right) - \pi^{-1}(x, D_A(x)). \tag{6}$$

Theoretically, we can compute the speed of a dynamic object, $v$, via the time interval $\Delta t$ of two consecutive frames:

$$v = \frac{\delta x_{A \to B}^{sf}}{\Delta t}. \tag{7}$$

*3.2. Geometric Segmentation*

It is not enough to select all dynamic outliers via optical flow information alone, as the optical flow can be greatly disturbed by the environment. Thus, we employ the epipolar geometry constraint between two consecutive frames to further check the dynamic keypoints. As the dynamic point is translated in the physical world, the distance between the corresponding feature point and the epipolar line should be larger than between those stationary points. Thus, we calculate the epipolar distance in $C_1 = (p_{d1} \ p_{d2} \ \cdots \ p_{dn})$ as the supplement for residual clustering. Figure 4 represents the geometric relationship between two consecutive frames.

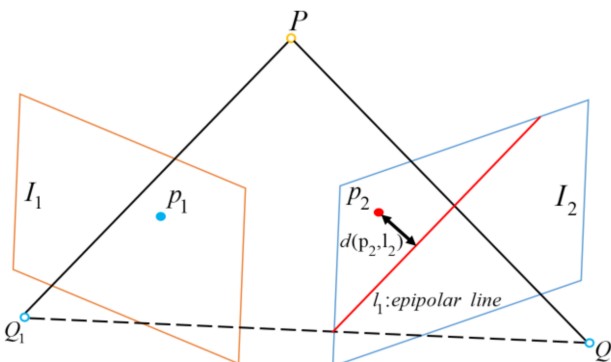

**Figure 4.** The epipolar geometry constraint: P represents the real-world point, while $p_1$ and $p_2$ are the matched keypoints where $d(p_2, l_2)$ indicates the distance between the keypoint and the epipolar line.

Let $l_{p_1}$ denote the epipolar equation:

$$l_{p_1} = \begin{bmatrix} X \\ Y \\ Z \end{bmatrix} = FP_1^T, \tag{8}$$

where $\begin{bmatrix} X & Y & Z \end{bmatrix}$ represents a line vector and F represents a fundamental matrix. Then, the epipolar distance is determined as follows:

$$d\left(p_2, l_{p_1}\right) = \frac{\left|P_2^T FP_1\right|}{\sqrt{\|X\|^2 + \|Y\|^2}}. \tag{9}$$

Since dynamic keypoints are not strictly subject to the epipolar geometry constraint, the longer the distance is, the greater the moving probability is. Based on this principle, we presume that the epipolar distance satisfies Gaussian distribution:

$$P\left(d_{p_2, l_{p1}}\right) = 1 - \frac{1}{\sqrt{2\pi}\sigma} \exp\left(-\frac{d\left(p_2, l_{p1}\right)^2}{2\sigma^2}\right), \tag{10}$$

where the parameter $\sigma$ represents the standard deviation of this distribution, which is set to 1 in our experiment. The mathematical expectation of the feature point distribution is set to 0 and $P\left(d_{p_2,l_{p1}}\right)$ represents the distance cost function.

Similarly, we designed an optical flow residual cost function to verify dynamic clustering to exclude the static keypoints that are mistakenly classified as dynamic clusters:

$$F(u_x, u_y) = \frac{1}{\exp\left(-\tau \bullet \left\| \delta x^{of}_{A \to B} \right\|\right) + 1},\tag{11}$$

where $F(u_x, u_y)$ is an optical flow residual cost function and $\tau$ is an impact factor, which is set to 0.3 in our system. $\left\| \delta x^{of}_{A \to B} \right\|$ is the L2 norm of the 2D optical flow residuals. To make the epipolar distance and optical flow residuals mutually beneficial, we introduce a cost function:

$$S = \alpha F + \beta P.\tag{12}$$

$\alpha$ and $\beta$ are the optical flow residuals impact factor and epipolar constraint impact factor, respectively, whose value setting should follow two ground rules:

i   Points that are clustered to the static background with large epipolar distances should be determined to be dynamic points through the joint cost function.

ii   Although some points are not in possession of epipolar distance, they have strong optical flow residuals. Theoretically, they are part of the geometry mask.

Thus, $\alpha$ and $\beta$ are set as 0.65 and 0.35, respectively, in our experiments.

$$\text{if } S > \kappa \text{ then, } p_i \in \vartheta_{dynamic}.\tag{13}$$

$\vartheta_{dynamic}$ is a keypoints container for the geometry mask, and the predefined parameter $\kappa$ reflects the system's sensitivity to the dynamic environment. We set different $\kappa$ thresholds in accordance with the semantic segmentation categories. Based on common sense, we are inclined to classify a person as a highly dynamic object and a chair as a static one; hence, we set $\kappa = 0.6$ for the person label, and $\kappa = 0.8$ for the chair label. Overall, we obtained the geometry mask through the combination of optical flow residual clustering and the epipolar check.

### 3.3. Semantic Segmentation

We adopted the Deeplabv3 network, operated on Tensorflow [33], to provide pixel-wise semantic information for common dynamic objects. This network, built with ResNet [34] as the backbone, includes four residual blocks and employs atrous convolution in cascade or in parallel to capture multi-scale context by adopting multiple atrous rates. The structure of this network is illustrated in Figure 5.

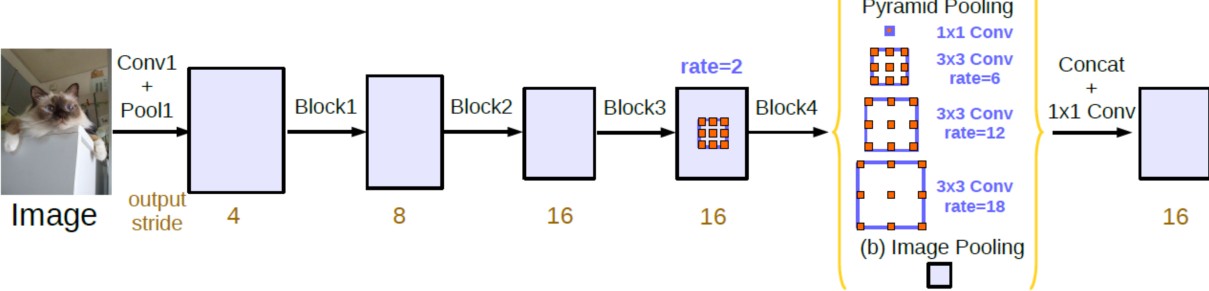

**Figure 5.** Structure of Deeplabv3.

Deeplabv3 can supply prior knowledge of moving objects, which is advantageous when detecting outliers. Potential dynamic instances such as people, bicycles, cars, motor-

cycles, airplanes, and buses are liable to appear in most complicated environments. This network is trained on the PASCAL VOC [35] dataset, which can segment 20 classes in total. If other classes are needed, the network can be finetuned with new training data. Raw RGB images constitute the input to Deeplabv3, while the output of the network is a binary mask for each object in the original frame. In our system, we considered that people often appear as moving objects (Figure 6 presents the semantic and geometric masks of moving person); therefore, when a certain proportion of dynamic feature points are located on a person, all the feature points on that person should be removed.

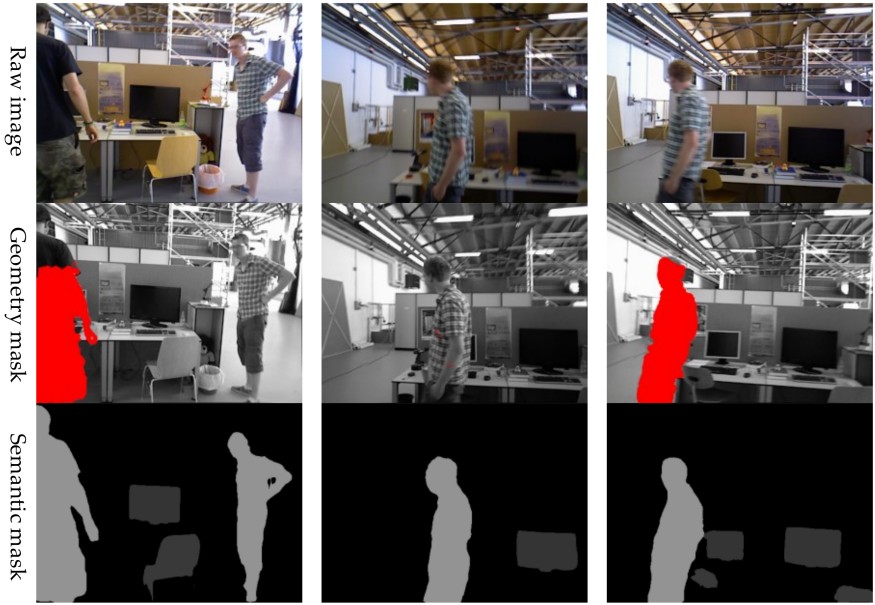

**Figure 6.** Examples of the results of geometric and semantic segmentation modules on a Technische Universität München (TUM) fr3/walking/xyz sequence. The first row presents the raw RGB images. The second and third rows present the results of the geometric and semantic segmentation modules, respectively. Of note, the second column indicates how the epipolar constraint can further identify dynamic points when the optical flow residuals become invalid.

*3.4. Outlier Rejection*

In this section, we present an outlier elimination mechanism to address the negative effect of moving objects. Based on common sense, we counted the feature points of the person and chair label respectively. When a certain proportion of dynamic points were detected, we considered the whole person or chair as a moving object. Particularly, we obtain a semantic instance object $M_{t \to t+1}^i$ ($i = 1$ means the person label and $i = 2$ means the chair label) from the spatial pixel domain $\Omega_p$. $r$ is the ratio used to determine whether the instance is static or dynamic. The motion judgement model was formulated as:

$$r_i = \frac{\sum\limits_{\Omega_p} \vartheta_{dynamic} \odot M_{t \to t+1}^i}{\sum\limits_{\Omega_p} M_{t \to t+1}^i} \, i = 1 \, , \, 2,$$ (14)

$$\prod \left( M_{t \to t+1}^i \right) = \left\{ \begin{array}{ll} 1 & r_i > r_{th} \\ 0 & r_i < r_{th} \end{array} \right. ,$$ (15)

where $\odot$ is an indicator function that shows the intersection of two elements, and $\prod$ is an indicator function, which equals 1 if the instance is in motion. $r_{th}$ is the threshold for the judgement model, which was set to 0.6 and 0.5 for the person and chair, respectively, in our experiment. The outliers rejection mechanism is shown in Algorithm 1 (using the person label to illustrate).

---

**Algorithm 1** Outliers Rejection Algorithm.

---

Input: dynamic keypoints container $\vartheta_{dynamic}$, instance object $M^i_{t \to t+1}$, current frame keypoints $P$
Output: Local feature map $M$
1:  for $p_i$ do
2:    if $p_i \in M^i_{t \to t+1}$ then
3:       Instance count $I\_cnt + +$
4:       if $p_i \in \vartheta_{dynamic}$ then
5:          Dynamic count $D\_cnt + +$
6:       end if
7:    end if
8:  end for
9:  if $D\_cnt / I\_cnt > 0.6$ then
10: Eraser keypoints $p_i$ in $M^i_{t \to t+1}$

---

## 4. Experiments

In this section, we evaluate the feasibility and effectiveness of our method using the public TUM [36] RGB-D dataset and our recorded dynamic office dataset, produced by the experimental quadcopter. The hybrid SLAM is integrated into the front end of ORB-SLAM2 and preprocesses the input image (see Algorithm 2).

---

**Algorithm 2** Dynamic keypoints filtering with original ORB-SLAM2 system.

---

Input: Image Sequence H, Depth Sequence D, Frames $(F_t \ F_{t+1})$
Output: Local feature frame $F_{output}$
1    for $(F_t \ F_{t+1})$ in H do
2       $\delta x^{ff}_{A \to B} = \text{Img\_pairs\_predict} \ (F_t \ F_{t+1})$ from PWC-Net
3       $M^i_{t \to t+1} = \text{Img\_predict} \ F_t$ from Deeplabv3
4       $T = solvePnPRansac(F_t, F_{t+1}, D, 50)$
5       Optimization: $T = BundleAdjustment(F_t, F_{t+1}, T)$
6       $\delta x^{of}_{t \to t+1} = CalcOpticalResiduals\left(\delta x^{ff}_{t \to t+1}, T\right)$
7       $C_1 = (p_{d1} \ p_{d2} \ \dots \ p_{dn}) = kmeans\left(\delta x^{of}_{t \to t+1}\right)$
8       $F\_M = FindFundamentalMatrix(F_t \ F_{t+1})$
9       for each matched pairs $(p_1, p_2)$ in $(F_t \ F_{t+1})$ do
10        $I_1 = FindEpipolarLine(p_1, F_M)$
11        $d = CalDistanFromEpipolarLine(p_2, I_1)$
12        $S = JointCostFunction(C_1, d)$
13        If $S > \kappa$ then
14           Append $p_i$ to $\vartheta_{dynamic}$
15        end if
16     end for
17   end for
18   Execute Outliers Rejection Algorithm
19   final

---

The TUM RGB-D dataset, captured by a Microsoft Kinect sensor, provides several sequences in dynamic environments with precise ground truth trajectories obtained via external motion capture equipment. Each sequence was recorded at a resolution of $640 \times 480$. The dynamic TUM sequences can be roughly divided into low dynamic (sitting) and highly dynamic (walking) sequences. In the low dynamic sequences, two men are sitting on chairs while talking and making gestures, but most of the body area remains static. By contrast, two pedestrians are walking in an office, where the moving area sometimes occupies most of the camera's field of view in the highly dynamic sequences. Moreover, there are four patterns of camera ego-motion in these datasets: moving along the xyz axes (xyz), rotating along the roll–pitch–yaw axes (rpy), remaining static (static), and moving along the circumference of a halfsphere with 1 m diameter (halfsphere). In addition to testing

on highly dynamic sequences, we also evaluated our proposed method in low dynamic sequences to demonstrate its feasibility.

We adopted the absolute trajectory error (ATE) and relative pose error (RPE) metrics to conduct a quantitative evaluation. ATE reflects the global consistency of the estimated trajectory compared with the ground truth, while RPE measures the translational and rotational drift of the visual odometry. We applied the widely-used root-mean-square error (RMSE, unit: m), median error (median), and standard deviation (Std) statistical analyses in this paper. RMSE is of particular importance as it can better indicate the stability and robustness of SLAM systems; therefore, different dynamic SLAM systems can be compared via their RMSE values. More specifically, we can calculate the improvement in our approach compared to the original ORB-SLAM2 via the RMSE. Let $\eta$ represent the improvement, $\beta$ denote the RMSE of the original SLAM, and $\alpha$ denote the value of our proposed method. Then, $\eta$ can be determined as follows:

$$\eta = \frac{\beta - \alpha}{\beta} \times 100\%. \tag{16}$$

### 4.1. Experiment on Public Datasets

We conducted a quantitative comparison evaluation (See Figure 7) on TUM datasets, the data from which are presented in Tables 1–3. With respect to the ATE metric in Table 1, RMSE values plummeted from 0.3948, as obtained by ORBSLAM2, to 0.0069, as achieved via our method. The positioning accuracy improved by up to 98.25%. In one typical sequence (fr3/w/xyz), the RMSE values also demonstrated a considerable decline from 0.6711 for ORBSLAM2, to 0.0152 for our robust hybrid SLAM. Similarly, for the rest of the dynamic TUM sequences, the improvement values in the proposed method were consistently close to 90%, while the downward trend in median and Std values is consistent with the RMSE. However, the improvements in the low dynamic sequences were less than those seen in the highly dynamic sequences. RMSE improvement values of 33.82%, 28.57%, and 24.88% were obtained for the low dynamic sequences. We hypothesize that these low improvement values are due to the ability of the original ORB-SLAM2 to easily handle outliers in low dynamic environments, so the room for improvement is limited.

**Table 1.** ATE (in meters) for our method and the original ORB-SLAM2.

| Category | Sequences | ORB-SLAM2 | | | Hybrid SLAM | | | Improvements | | |
|---|---|---|---|---|---|---|---|---|---|---|
| | | Median | RMSE | Std | Median | RMSE | Std | Median | RMSE | Std |
| Low dynamic sequence | fr3/s/xyz | 0.0068 | 0.0085 | 0.0043 | 0.0078 | 0.0100 | 0.0050 | −14.71% | −17.65% | −16.28% |
| | fr3/s/static | 0.0067 | 0.0084 | 0.0039 | 0.0046 | 0.0060 | 0.0030 | 31.34% | 28.57% | 23.08% |
| | fr3/s/rpy | 0.0132 | 0.0213 | 0.0130 | 0.0115 | 0.0160 | 0.0085 | 12.88% | 24.88% | 34.62% |
| | fr3/s/halfsphere | 0.0136 | 0.0207 | 0.0132 | 0.0113 | 0.0137 | 0.0061 | 16.91% | 33.82% | 53.79% |
| High dynamic sequence | fr3/w/xyz | 0.4598 | 0.6711 | 0.3752 | 0.0114 | 0.0152 | 0.0075 | 97.52% | 97.74% | 98.00% |
| | fr3/w/static | 0.2812 | 0.3948 | 0.1650 | 0.0052 | 0.0069 | 0.0034 | 98.15% | 98.25% | 97.94% |
| | fr3/w/rpy | 0.5655 | 0.7005 | 0.2866 | 0.0781 | 0.1050 | 0.0504 | 86.19% | 85.01% | 82.41% |
| | fr3/w/halfsphere | 0.3512 | 0.4320 | 0.1764 | 0.0208 | 0.0280 | 0.0144 | 94.08% | 93.52% | 91.84% |

**Table 2.** Translational drift of RPE (in meters) for our method and the original ORB-SLAM2.

| Category | Sequences | ORB-SLAM2 | | | Hybrid SLAM | | | Improvements | | |
|---|---|---|---|---|---|---|---|---|---|---|
| | | Median | RMSE | Std | Median | RMSE | Std | Median | RMSE | Std |
| Low dynamic sequence | fr3/s/xyz | 0.0064 | 0.0082 | 0.0041 | 0.0064 | 0.0086 | 0.0046 | 0.00% | −4.88% | −12.20% |
| | fr3/s/static | 0.0040 | 0.0056 | 0.0029 | 0.0033 | 0.0045 | 0.0024 | 17.50% | 19.64% | 17.24% |
| | fr3/s/rpy | 0.0076 | 0.0126 | 0.0082 | 0.0072 | 0.0105 | 0.0060 | 5.26% | 16.67% | 26.83% |
| | fr3/s/halfsphere | 0.0060 | 0.0082 | 0.0045 | 0.0068 | 0.0090 | 0.0047 | −13.33% | −9.76% | −4.44% |
| High dynamic sequence | fr3/w/xyz | 0.0176 | 0.2726 | 0.0164 | 0.0077 | 0.0113 | 0.0064 | 56.25% | 95.85% | 60.98% |
| | fr3/w/static | 0.0053 | 0.0137 | 0.1074 | 0.0037 | 0.0056 | 0.0033 | 30.19% | 59.12% | 96.93% |
| | fr3/w/rpy | 0.0162 | 0.0493 | 0.0437 | 0.0126 | 0.0237 | 0.0162 | 22.22% | 51.93% | 62.93% |
| | fr3/w/halfsphere | 0.0136 | 0.0372 | 0.0320 | 0.0084 | 0.0135 | 0.0083 | 38.24% | 63.71% | 74.06% |

**Table 3.** Rotational drift of RPE (in meters) for our method and the original ORB-SLAM2.

| Category | Sequences | ORB-SLAM2 | | | Hybrid SLAM | | | Improvements | | |
|---|---|---|---|---|---|---|---|---|---|---|
| | | Median | RMSE | Std | Median | RMSE | Std | Median | RMSE | Std |
| Low dynamic sequence | fr3/s/xyz | 0.0058 | 0.0079 | 0.0041 | 0.0058 | 0.0077 | 0.0040 | 0.00% | 2.53% | 2.44% |
| | fr3/s/static | 0.0030 | 0.0041 | 0.0021 | 0.0025 | 0.0037 | 0.0020 | 16.67% | 9.76% | 4.76% |
| | fr3/s/rpy | 0.0071 | 0.0099 | 0.0056 | 0.0071 | 0.0092 | 0.0045 | 0.00% | 7.07% | 19.64% |
| | fr3/s/halfsphere | 0.0065 | 0.0089 | 0.0046 | 0.0067 | 0.0089 | 0.0045 | −3.08% | 0.00% | 2.17% |
| High dynamic sequence | fr3/w/xyz | 0.0100 | 0.0157 | 0.0096 | 0.0057 | 0.0095 | 0.0067 | 43.00% | 39.49% | 30.21% |
| | fr3/w/static | 0.0040 | 0.0070 | 0.0047 | 0.0031 | 0.0041 | 0.0022 | 22.50% | 41.43% | 53.19% |
| | fr3/w/rpy | 0.0107 | 0.0196 | 0.0143 | 0.0085 | 0.0135 | 0.0082 | 20.56% | 31.12% | 42.66% |
| | fr3/w/halfsphere | 0.1001 | 0.0240 | 0.0204 | 0.0072 | 0.0098 | 0.0052 | 92.81% | 59.17% | 74.51% |

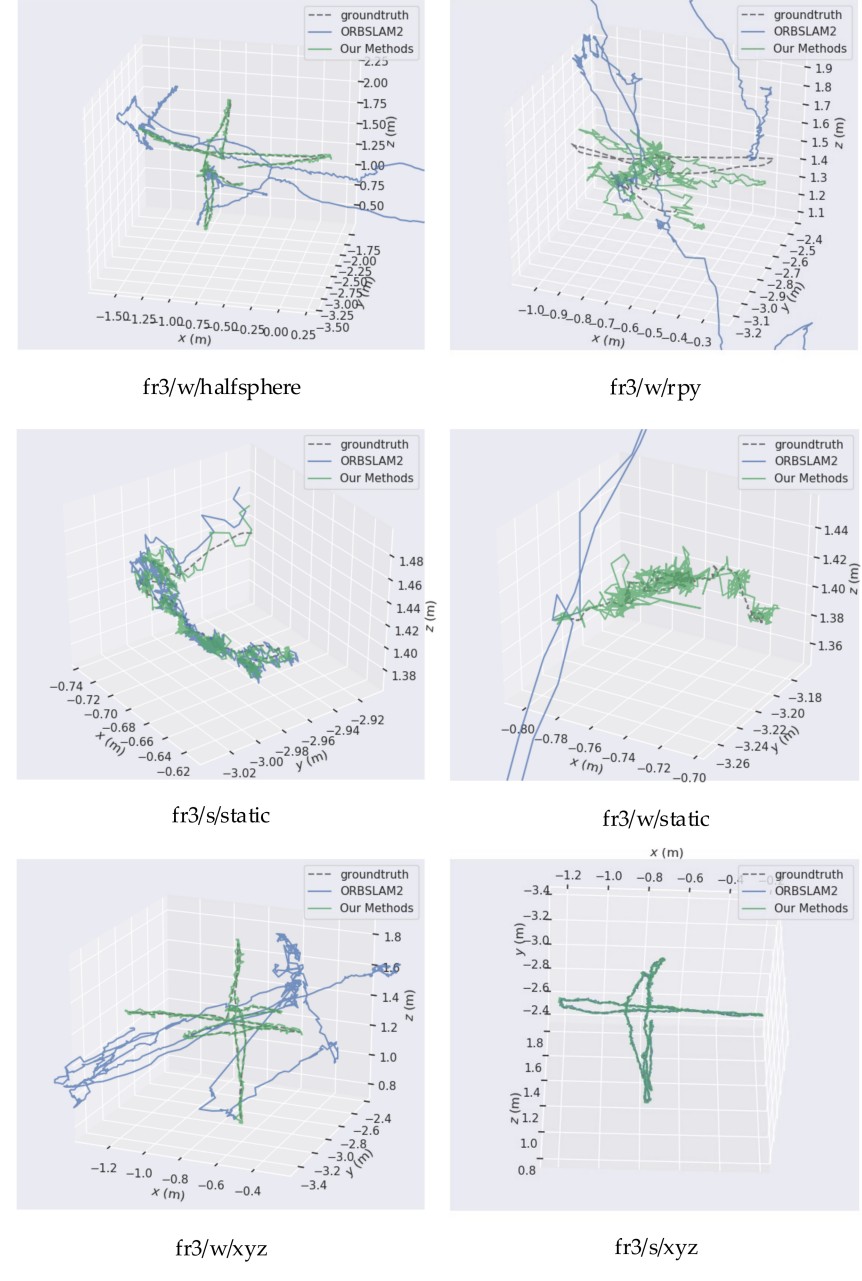

**Figure 7.** Results of the fr3/w/halfsphere, fr3/w/rpy, fr3/s/static, fr3/w/static, fr3/w/xyz, and fr3/s/xyz sequences for the TUM dynamic RGB-D datasets. We compared the trajectories estimated by our methods and by the original ORB-SLAM2 with the ground truth.

From Tables 2 and 3, our proposed method functions better than ORBSLAM2 in terms of translational drift; RMSE values decreased from 0.2726 to 0.0113 for fr3/w/xyz sequences—an improvement of around 95.85%. The other three improvements seen in highly dynamic sequences level off at approximately 60%. With regard to the rotational drift of RPE, the ORBSLAM2 RMSE values were 0.0157, 0.0070, 0.0196, and 0.0240, while the results obtained by our method were 0.0095, 0.0041, 0.0135, and 0.0098. Taking the above data together, the hybrid SLAM improvement values varied from 39.49% to 59.17% across all highly dynamic sequences. All in all, the emulation strongly supports the fact that our hybrid SLAM outperforms ORBSLAM2.

From the contrasting results seen between translational and rotational drift, we concluded that dynamic objects affect the translation accuracy more than the rotation accuracy. We attribute this difference to two main factors: (a) rotational drift can also trigger translational drift, so translational drift is generally larger than rotational drift and the room for improvement is larger; and (b) in the TUM sequences, dynamic movement primarily appears in the form of translational movement. This form of movement has a more powerful effect on translational measurement than on rotational measurement. Therefore, translational improvement is naturally more prominent after rejecting the negative outliers.

Additionally, we discovered that our method underperformed in one low dynamic sequence: fr3/sitting/xyz. The negative results seen here may be attributed to the fact that most of the area of the person label is static, but we mistakenly erased the keypoints in that area. In other words, our fixed threshold $\kappa$ (Equation (13)) is well adapted to highly dynamic sequences rather than low dynamic sequences, which led to poor performance—due to the shortage of sufficient keypoints—in this specific sequence. Theoretically, our system is projected to function better by setting a variable threshold $\kappa$ to conform to a wide variety of movement scenarios in scenes.

### 4.2. Comparison with Other Dynamic SLAM Systems

We also conducted a comparative test between our system and other SLAM systems designed for dynamic scenarios to further determine the effectiveness of our approach. Learning-based SLAM systems—such as Dyna-SLAM [18] and DS-SLAM [17]— and geometry-based SLAM systems—such as Static Point Weighting-SLAM [15]—were included for contrast.

The line chart presented in Figure 8 provides information regarding the disparate dynamic SLAM results from tests on benchmark datasets in terms of ATE. Overall, for all highly dynamic sequences, the hybrid SLAM RMSE (Unit: m) values were the lowest, being 0.0152, 0.0280, 0.0069, and 0.1050. For the semantic association methods, hybrid SLAM RMSE metrics leveled off below 0.05, similarly to DS-SLAM and Dyna-SLAM. However, when encountering the very harsh fr3/w/rpy sequence, RMSE results drastically declined from 0.4442 in DS-SLAM, to 0.1050 in hybrid SLAM—a drop of approximately four times, implying the superior intelligence of our approach. Overall, the chart illustrates, to a large extent, our system's superiority to the other systems on TUM dynamic datasets.

### 4.3. Robustness Test in Real Environments

We conducted further experiments with a live RGB-D camera attached to the quadcopter. As can be seen in Figure 9, our system isolates dynamic keypoints from detected ones. In our experiment, a person walked in front of the camera while the quadcopter hovered (see Figure 10). The red keypoints represent outliers, as determined by our proposed approach, while the yellow keypoints indicate the keypoints that will be fed into our system to estimate the camera position. In addition, the speed of the dynamic objects can be computed by leveraging the optical flow residuals. As a result, the average estimated speed of the pedestrian was 0.8 m/s (from measurements of 0.76, 1.07, and 0.89 m/s), which was very close to the true value.

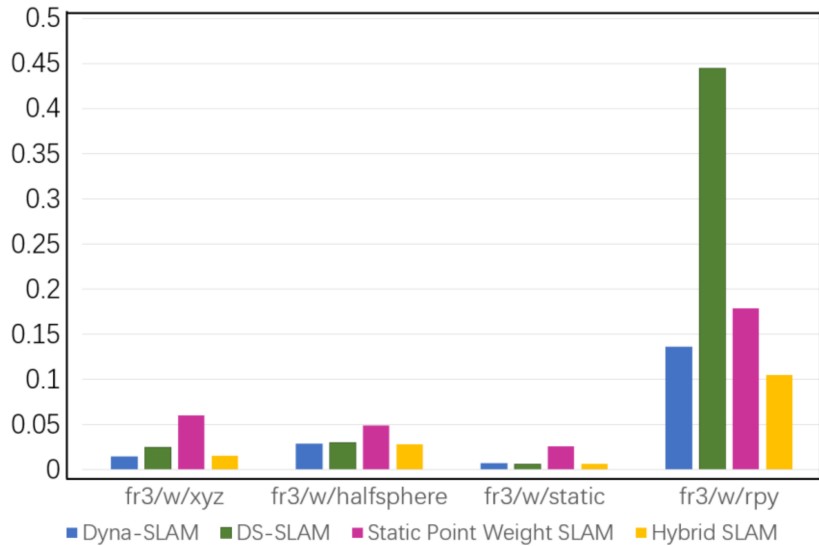

**Figure 8.** ATE comparison results.

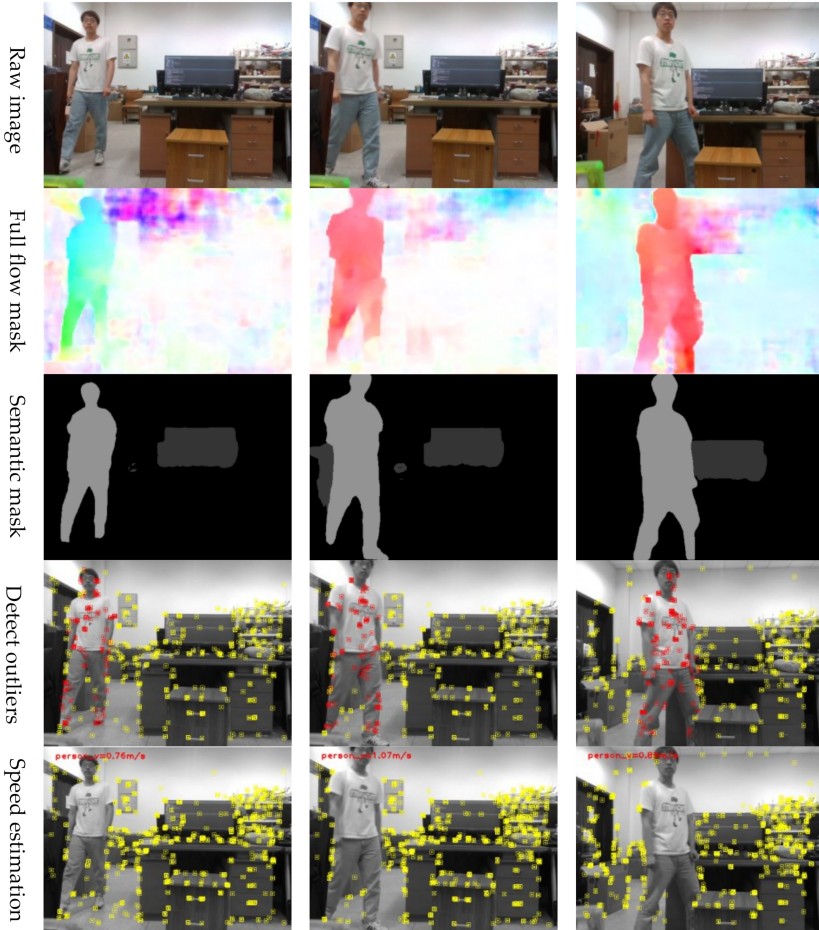

**Figure 9.** Real environment experiments conducted with a live RGB-D camera under physical conditions. The first row presents the raw RGB images and the second row shows the full optical flow as predicted by PWC-Net. The third row is the semantic mask, and the fourth row indicates that our system can separate the dynamic keypoints from real-world conditions. The final row demonstrates the determination of the speed of dynamic objects after the removal of outliers.

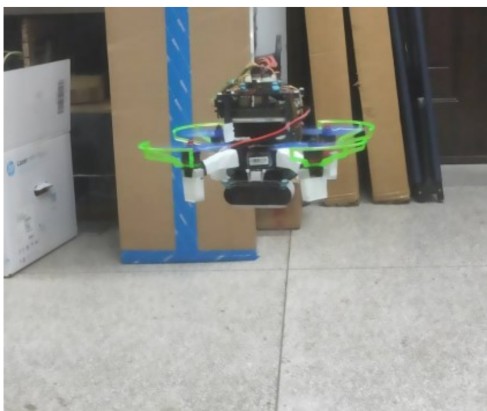

**Figure 10.** The quadcopter records the datasets in dynamic environments.

## 5. Conclusions

In this paper, we propose a novel approach to the elimination of dynamic points, which can greatly improve the feasibility and effectiveness of the state-of-the-art ORB-SLAM2 system. With the help of the two deep learning networks, PWC-Net and Deeplabv3, we can achieve geometric and semantic segmentation of the raw image. Moreover, we developed an outliers rejection strategy to allow our system to easily deal with dynamic environments. Our scheme can determine the speed of a common dynamic instance, which is highly advantageous for robots tracking dynamic objects. We conducted experiments on public TUM datasets coupled with physical conditions to verify the feasibility of our approach. Quantitative evaluations demonstrated that our method could greatly improve the performance of the original ORB-SLAM2 system. We also demonstrated that our system outperforms other published dynamic systems on TUM dynamic datasets.

In future work, the fixed threshold $\kappa$ will be discarded on account of its low robustness. As a consequence, we expect to introduce an adjustable threshold $\kappa$ that varies with the environment to further accelerate deployment under physical conditions. In other words, $\kappa$ is estimated to increase in low dynamic scenarios to obtain adequate static keypoints, and slide in highly dynamic scenarios to forsake dynamic keypoints. We will also consider applying reinforcement learning, an emerging field of artificial intelligence (AI) to this subject.

Additionally, the semantic map can be used for more advanced human–computer interaction tasks in field of vision SLAM. Instead of a linear combination of the optical flow residual and epipolar distance, we may consider constructing a learning-based nonlinear energy function to distinguish the dynamic keypoints.

**Author Contributions:** Methodology, Changze Li; formal analysis, Sheng Miao; investigation, Dazheng Wei; writing—original draft preparation, Sheng Miao; writing—review and editing, Changze Li; supervision, Xiaoxiong Liu. All authors have read and agreed to the published version of the manuscript.

**Funding:** This research was funded by the National Natural Science Foundation of China, grant number 61573286, and the Aeronautical Science Foundation of China, grant number 201905053003.

**Informed Consent Statement:** Informed consent was obtained from all subjects involved in the study.

**Data Availability Statement:** Publicly available datasets were analyzed in this study. This data can be found here: http://vision.in.tum.de/data/datasets/rgbd-dataset (accessed on 30 June 2021).

**Acknowledgments:** The authors would like to thank the editor, associate editor, and anonymous reviewers for processing our manuscript.

**Conflicts of Interest:** The authors declare no conflict of interest.

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
