# Peer review of "A Visual SLAM Robust against Dynamic Objects Based on Hybrid Semantic-Geometry Information"

_ijgi, doi:10.3390/ijgi10100673_

Round 1
Reviewer 1 Report
The paper proposes a hybrid SLAM based system which exploits semantic-geometry information to reduce rotational and horizontal errors.
The idea proposed on the article is not novel itself (see Chao Yu et al 2018, Nikolay Atanasov 2018) as in the literature semantic association to visual SLAM already proved to get better results.
Notwithstanding, this paper has merit, as it makes experiments on both proprietary and benchmark datasets. Moreover, the practical implication of using depth information for semantic scene understanding deserves attention by the research community. So, I support this publication, but I can reconsider the manuscript only after the following aspects will be clarified by the authors:
Points 1, 2 and 3 in the introduction should be revised, as these claims do not meet the specific findings along with the manuscript. Deeplabv3 is not mentioned, introducing the semantics on prediction network is not novel and providing the speed of common dynamic instances is only a technical implementation. Please condense the findings in one more focused real contribution, as done in the paper.
Not all the readers are familiar with visual SLAM, I suggest to expand the literature with a dedicated section where the authors might highlight the main difference of their approach with the state of the art methods.
For instance, ORB-SLAM is only one way to approach the problem, but there are other frameworks and libraries that proved their effectiveness that are here completely neglected. Please add a comparison with the existing literature.
Line 339 “but we erase the keypoints in that area”.
Keypoints removal has been done manually? how the threshold was set? Shouldn't AI help in this? This aspect was repeated in the future work section by the authors, but this aspect should be more clarified
In table 3 those values that goes to Zero are strange to me, please double check.
The technical aspect of the DeepLabV3 are completely neglected, I suggest to add more information about the network performances, how was it trained, does the method perform back and forward propagation, how it affects the loss?
The differences between rotational and translational drift are really high and this aspect should be expanded in the discussion
Reviewer 2 Report
I love this paper because it finally proposes a fast-performing approach to dynamic points in SLAM. Introduction positions very well the contribution of the paper, as well the conclusions do. Ideally it should open a new research perspective for SLAM-based robots. Both mathematical model and experiments are well illustrated.
The text is conceptually clean, but it plagued by clumsy phrasing, typos, and grammar errors, which, altogether, make reading harder that it should be. Let me give an example. In row 29 authors write “employ camera as the external sensors to get the accurate position in some sophisticated environments recent years” instead of “in recent years, use cameras as external sensors for accurate positioning in complex environments”. I strongly recommend that Authors review ALL the text.
Figures, though well explained, are not always accurate. Elements of Figure 1 are small and hardly readable. The same holds for Figure 2, Figure 4, Figure 7.
Author Response
请参阅附件。

Reviewer 3 Report
Interesting paper, where especially the Methodology part was well-written. The end result is rather mature and sources hopefully become available to the research community later on.
Some comments:
Please improve the mathematical typography, especially:
-choose either l_{p1} or l_{p_1}. I think the latter is recpommendable, since Fig. 2 has p_1 and p_2. Fix in every occurrence.
-change \text{u}_x to u_x in Eq. 11 and the following "...where" sentence
-add "," in the end of Eq. 11 and change "Where" to "where"
-Eq 13 should be \text{if } S > \kappa, \text{ then } P_i \in ... (have words in text format, not in italics
-p.8 Fig. 5 may require some more formatting according to the editorial principles. Using labels for each sub img could be an overkill, but let editor decide over it.
-change "realword" --> "real world" everywhere
Building the Methodology presentation on Eqs. 2,4 and 6 was an elegant choice.
Author Response
请参阅附件。

Reviewer 4 Report
Article review “A visual SLAM against dynamic objects based on hybrid 2 semantic-geometry information” – authors proposed a novel approach to detect and eliminate dynamic points, which can improve the feasibility and effectiveness of state-of-the-art Simultaneous Localization and Mapping (SLAM) “ORB-SLAM2” algorithm and they used two deep learning network PWC-Net and Deeplabv3, to obtain the geometric segmentation and semantic segmentation of the raw image. The authors showed that they improved the algorithm and got better results with their improved algorithm.
Individual remarks:
- It is suggested that the authors better and more comprehensively describe the obtained results, especially in which their algorithm is better than others, especially than ORB-SLAM2.
- The conclusion should be better and more comprehensively written and it should be emphasized what the author's improvements are and how much better they got the results.
- Figures 1, 5, 6 and 8 - low resolution and image size, names cannot be read, I suggest that the images be larger - on the figures 5 and 8 I suggest leaving only three examples instead of five but to be larger and with a higher resolution. Figure 8 should also be shown larger – ground truth the trajectory is not visible; original ORBISLAM2 trajectory and new proposed method should be shown in such colors and line thicknesses to see the difference - numbers and letters are poorly legible/illegible.
- Algorithm 1 and 2 low resolution – we suggest authors to write write algorithms rather than using screen shout.
- Line 249 and 341 - equalize font and dimensions.
Author Response
请参阅附件。

This manuscript is a resubmission of an earlier submission. The following is a list of the peer review reports and author responses from that submission.